



**Soil erodibility estimation by using five methods of estimating *K* value: A case study in Ansai watershed of**
**Loess Plateau, China**
**Wenwu Zhao, Hui Wei, Lizhi Jia, Stefani Daryanto, Yanxu Liu**
State Key Laboratory of Earth Surface Processes and Resources Ecology, Faculty of Geographical Science,
Beijing Normal University, Beijing 100875, China
Institute of Land Surface System and Sustainable Development, Faculty of Geographical Science, Beijing
Normal University, Beijing 100875, China
*Correspondence to*: Wenwu Zhao (Zhaoww@bnu.edu.cn)
**Abstract**
The objectives of this work were to select the possible best texture-based method to estimate K and
understand possible indirect environmental factors of soil erodibility. In this study, 151 soil samples were
collected during soil surveys in Ansai watershed. Five methods of estimating *K* value were used to estimate soil
erodibility, including the erosion-productivity impact model (EPIC), the nomograph equation (NOMO), the
modified nomograph equation (M-NOMO), the Torri model and the Shirazi model. The *K* values in Ansai
watershed ranged between 0.009 and 0.092 t $hm^2$ hr/(MJ mm $hm^2$). The *K* values based on Torri, NOMO, and
Shirazi models were similar and were located close to each other in the Taylor diagrams. By combining the
measured soil erodibility, we suggested Shirazi and Torri model as the optimal models for Ansai watershed. The
correlations between soil erodibility and the selected environmental variables changed for different vegetation
type. For native grasslands, soil erodibility had significant correlations with terrain factors. For most artificially
managed vegetation types (e.g., apple orchards) and artificially restored vegetation types (e.g., sea buckthorn),
the soil erodibility had significant correlations with the growing conditions of vegetation. The dominant factors
that influenced soil erodibility differed with different vegetation types. Soil erodibility had indirect relationship





with not only environmental factors (e.g., elevation and slope), but also human activities which potentially
altered soil erodibility.
**Keywords:** Influencing factors, Soil erodibility, Variation features, Shirazi model, Torri model
**1 Introduction**
Soil erodibility ($K$), as one of the key factors of soil erosion (Igwe, 2003; Fu et al., 2005; Ferreira et al.,
2015), is defined as the susceptibility of soil to erosional processes (Bagarello et al., 2012; Bryan et al., 1989). It
has been extensively used in both theoretical and practical approaches to measure soil erosion. Yet it is a
complex concept and is affected by many factors, including soil properties (e.g., soil texture, permeability and
structural stability) (Chen et al., 2013; Wang et al., 2015; Manmohan et al., 2012); terrain (Wang et al., 2012;
Mwaniki et al., 2015; Parajuli et al., 2015); climate (Hussein et al., 2013; Sanchis et al., 2010); vegetation
(Sepúlveda-Lozada et al., 2009); and land use (Cerdà et al., 1998; Tang et al., 2016). In order to calculate soil
erodibility, many strategies have been used to perform research to understand soil erodibility, including
measurements of physical and chemical soil properties, instrumental measurements, mathematical models and
graphical methods (Wei et al., 2017). Although a direct measurement of soil erosion with large plots under
natural rainfall over long-term period can provide more accurate estimates of soil erodibility, this method is time
consuming and very expensive (Bonilla et al., 2012; Vaezi et al., 2016a, b). Therefore, mathematical models are
more commonly used to estimate soil erodibility.
Some of the most common estimation models are the nomogram model and the modified nomogram model,
which were established by Wischmeier (Wischmeier et al., 1971, 1978); the erosion-productivity impact model
(EPIC), which was developed by Williams (Williams et al., 1990); the best nonlinear fitting formula using the
physical and chemical properties of the soil, which was developed by Torri (Torri et al., 1997); and the
estimation model developed by Shirazi that is using the average size of the soil geometry (Shirazi et al., 1988).



Each estimation method may differ in terms of their applicability, even within the same area because different
estimation methods include different physical and chemical soil properties (Lin et al., 2017; Wang et al., 2013b;
Kiani et al., 2016). Consequently, the estimated results can differ significantly because soil conditions vary by
region (Lin et al., 2017; Wang et al., 2013b). Selecting the optimal estimation method of soil erodibility is
therefore critical to estimate the amount of soil erosion.
Soil erosion in the Loess Plateau of China is among the highest in the world (Fu et al., 2009; Huang et al.,
2016). The area affected by soil and water loss is as large as $4.5 \times 10^5$ km$^2$ (~71% of the local land area) and the
long-term average sediment loss is up to $1.6 \times 10^9$ t (Fu et al., 2017). To maintain water quality and to control soil
erosion (Fu et al., 2011), the Chinese government has implemented a large-scale policy to convert farmlands to
forests and grasslands since the 20th century (Lü et al., 2012; Feng et al., 2013b; Wu et al., 2016). Although this
large-scale introduction of vegetation should reduce soil erosion, the extent of the reduction remains unclear.
Accordingly, different estimation methods should be used to calculate erosion factors, including soil erodibility
factor. In this article, Ansai watershed in Loess Plateau of China was chosen as a case study, and the above five
estimation methods of estimating $K$ value were used, and the objectives of this study are (1) to estimate soil
erodibility factor with different methods; (2) to select the possible best texture-based method to estimate $K$; (3) to
understand possible indirect environmental factors on soil erodibility.
**2 Materials and methods**
**2.1 Study area**
The Ansai watershed (108°5′44″-109°26′18″E, 36°30′45″-37°19′3″N) is located in the upper reaches of the
Yanhe River. This watershed lies in the northern part of Shanxi province and the inland hinterland of the
northwestern Loess Plateau and at the edge of the Ordos basin. It belongs to the typical loess hilly-gully region
and covers an area of approximately 1334 km$^2$. The topography is complex and varied, and the ground surface is





fragmented. The elevations within the watershed are high in the northwest and low in the southeast, and these
elevations range from 997 to 1731 m above sea level. The watershed belongs to the mid-temperate continental
semi-arid monsoon climate region. The average annual precipitation is 505.3 mm, and 74 percent of the rainfall
occurs from June to September. The predominant land use types in the Ansai watershed are rain-fed farmland,
apple orchard, native grassland, pasture grassland, shrubland, and forest (Feng et al., 2013a). The soil type in this
study area is loess soil with low fertility and high vulnerability to erosion (Zhao et al., 2012; Yu et al., 2015).
**2.2 Sample point setting**

The soil data used in this study came from 151 typical sample data sets that were obtained during soil

surveys conducted from July to September in 2014. The soil types of all 151 sample points are loess soil.
Representative vegetation were selected, which included (1) natural vegetation, including native grassland (NG);
(2) artificially managed vegetation types, including apple orchards (AO) and farmland (FL); and (3) artificially
restored vegetation types, including pasture grassland (PG), sea buckthorn (SB), *Caragana korshinskii* (CK),
David's peach (DP), and black locust (BL). The distance between each vegetation sampling site was at least 2
km, the area of each vegetation type was greater than 30 m by 30 m, and the selected sample plots were
distributed evenly within the study area. The sample plots within the farmland and grassland had a size of 2 m by
2 m, whereas the corresponding dimensions for the sample plots within the shrubland and forest areas were 5 m
by 5 m and 10 m by 10 m, respectively. Each sample plot was repeated three times. The locations of the
sampling points were determined using a GPS unit (Garmin eTrex 309X). The collected soil samples were taken
back to the laboratory, dried naturally, ground and filtered with a 2-mm sieve. The grain size distributions of the
soil samples were evaluated using the hydrometer method. The size classes of the particles in this study were as
follows: sand (0.005-2.0 mm), silt (0.002-0.05 mm) and clay (< 0.002 mm).

To fully explore the primary factors influencing soil erodibility in the Ansai watershed, we chose four types





of environmental factors, including physicochemical soil properties, topographic factors, climate factors and
vegetation factors. While soil erodiblity does not directly depend on environmental factors, soil properties such
as soil particle and soil organic matter can be affected by environmental factors. Soil erodibility thus has indirect
relationship with the environmental factors. These environmental factors covered 20 independent variables,
specifically elevation (Ele), slope position (SP), slope aspect (SA), slope gradient (SG), slope shape (SS), clay
(Cla) content, silt (Sil) content, sand (San) content, organic matter (OM) content, soil bulk density (SBD),
porosity (Por), average annual rainfall (AAR), vegetation coverage (VC), aboveground biomass (AB), vegetation
height (VH), litter biomass (LB), plant density (PD), crown (Cro), basal diameter (BD), and branch number (BN).
All of the environmental factors were derived from the field surveys. The main characteristics and sampling
numbers for the study area are shown in Table 1, and the sampling points are shown in Fig.1. Based on the
results of the Spearman correlation analysis, we then retained some environmental variables that displayed
significant correlations ($P < 0.05$) with soil erodibility to perform a principal component analysis (PCA) and to
obtain the minimum data set (MDS) (Xu et al., 2008). Only principal components (PCs) with eigenvalues $N >$
1.0 and only variables with highly weighted factor loadings (i.e., those with absolute values within 10% of the
highest value) were retained for the MDS (Mandal et al., 2008).
**2.3 Research methods**

Soil erodibility indicates the degree of difficulty that soil becomes separated, eroded and transported by

rainfall erosion (Wang et al., 2013a; Cerdàet al., 2017). Soil erodibility factor, which is commonly known as the
*K*-factor in the model, is defined as the average rate of soil loss per unit of rainfall erosivity index from a
cultivated continuous fallow plot on a 22.1-m-long, 9% slope in the universal soil loss equation (Zhang et al.,
2008). To minimize bias from using only one estimation method, we estimated the *K* values using five estimation
models (i.e., EPIC, NOMO, M-NOMO, Torri and Shirazi), that have been widely applied in the research on soil





erodibility (Wischmeier et al., 1971, 1978; Williams et al., 1990; Torri et al., 1997; Shirazi et al., 1988).
*2.3.1 K value estimation using the EPIC model*
The erosion-productivity impact model (EPIC) developed by Williams (Williams et al. 1990) is as follows:

$$K = [0.2 + 0.3e^{-0.0256\,SAN\left(1-\frac{SIL}{100}\right)}]\,\left(\frac{SIL}{CLA+SIL}\right)^{0.3}(1.0 - \frac{0.25C}{C+e^{3.72-2.95C}})\,(1.0 - \frac{0.7SN_1}{SN_1 + e^{-5.51+22.9SN_1}}) \tag{1}$$

where *SAN* is the percent sand content, *SIL* is the percent silt content, *CLA* is the percent clay content, *C* is the
percent organic carbon content, and $SN_1$ = 1-*SAN*/100. The resulting *K* value is reported in United States
customary units of [short ton ac h / (100 ft short ton ac in)].
*2.3.2 K value estimation using the NOMO model*
Wischmeier (Wischmeier et al., 1971) proposed this model after analyzing the relationship between soil
erosion and five soil characteristic indicators, including the percent silt+very fine sand fraction (0.05-0.1 mm),
the percent sand fraction, the soil organic matter content, a code for soil structure, and a code for soil
permeability:

$$K = [2.1 \times 10^{-4} M^{1.14}(12 - OM) + 3.25(S - 2) + 2.5(P - 3)] / 100 \tag{2}$$

where *M* is the product of the percent of silt+very fine sand and the percent of all soil fractions other than clay,
*OM* is the soil organic matter content (%), *S* is the soil structure code, and *P* is the soil permeability code. The
resulting K value is reported in United States customary units of [short ton ac h/(100 ft short ton ac in)].
*2.3.3 K value estimation using the M-NOMO model*
On the basis of the universal soil loss equation (USLE) model, the RUSLE model was modified for
calculating soil erodibility; that is, a revised nomograph equation was devised (Wischmeier et al., 1978) based on
the nomograph equation. The revised nomograph equation is:

$$K = [2.1 \times 10^{-4} M^{1.14}(12 - OM) + 3.25(2 - S) + 2.5(P - 3)] / 100 \tag{3}$$

where *M* is the product of the percent of silt+very fine sand and the percent of all soil fractions other than clay,



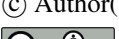

*OM* is the soil organic matter content (%), *S* is the soil structure code, and *P* is the soil permeability code. The
resulting K value is reported in United States customary units of [short ton ac h/(100 ft short ton ac in)].
*2.3.4 K value estimation using the Torri model*
Torri (Torri et al., 1997) established this model in 1997 using data describing soil particle size and soil
organic matter content. The model has few parameters, and acquisition of the relevant data is simple. The
formula used in evaluating this model is as follows:

$$K = 0.0293(0.65 - D_g + 0.24 D_g{}^2) \times \exp\left\{-0.0021\frac{OM}{c} - 0.00037\left(\frac{OM}{c}\right)^2 - 4.02c + 1.72c^2\right\} \tag{4}$$

where *OM* is the percent content of soil organic matter, and *c* is the percent content of clay. In addition, the $D_g$
can be calculated by the following formula:

$$D_g = \sum f_i \lg \sqrt{d_i d_{i-1}} \tag{5}$$

where $D_g$ is the Napierian logarithm of the geometric mean of the particle size distribution, $d_i$ (mm) is the
maximum diameter of the *i*-th class, $d_{i-1}$ (mm) is the minimum diameter and $f_i$ is the mass fraction of the
corresponding particle size class. We calculate the $D_g$ based on three particle size classes, namely sand, silt, and
clay. The resulting K values are reported in the international units of [(t hm² h)/(MJ mm hm²)].
*2.3.5 K value estimation using the Shirazi model*
Shirazi (Shirazi et al., 1988) put forward a model that is appropriate for situations involving fewer physical
and chemical properties of the soil materials. He suggested that K values can be calculated through considering
only the geometric mean diameter ($D_g$) of the soil grains. The relevant formula is:

$$K = 7.594\left\{0.0034 + 0.0405\, e^{-\frac{1}{2}\left[\frac{\log(D_g)+1.659}{0.7101}\right]^2}\right\} \tag{6}$$

$$D_g(mm) = e^{0.01\sum f_i \ln m_i} \tag{7}$$

where $D_g$ is the geometric mean diameter of the soil particles, $f_i$ is the weight percentage of the *i*-th particle size



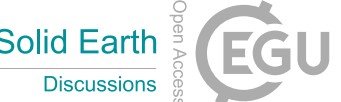

fraction (%), $m_i$ is the arithmetic mean of the particle size limits for the $i$-th fraction (mm), and $n$ is the number of
particle size fractions. The resulting $K$ value is reported in United States customary units of [short ton ac h/(100
ft short ton ac in)].

To increase the comparability of the results from the different estimation models, our research adopted the

international units for the $K$ values, [t hm² hr/(MJ mm hm²)]. The international $K$ value is equal to the $K$ value
reported in the United States customary units, multiplied by 0.1317.

To clarify the form of the distribution, we adopted the Kolmogorov-Smirnov test (Table 2) and made the

frequency distribution figures of soil erodibility for each model (Fig. 2). The $P$ value>0.05 showed that the $K$
values obtained using the five methods were normally distributed. Therefore, the soil erodibility $K$ values
measured within the study area can be analyzed directly using statistical methods without data conversion (Fang
et al. 2016).
*2.3.6 K value* comparisons

In order to discuss the possible best texture-based method to estimate $K$, related researches on K estimation,

especially the measured value of K in Loess Plateau of China, have been collected. Taylor Diagram was also
used to compare the difference between models.
**3 Results**
**3.1 Soil erodibility based on five different models in Ansai watershed**

We found that the descriptive statistics of the K values in Ansai watershed differed when different models

were used (Table 2). The range of $K$ values based on the five methods were between 0.032 and 0.060, 0.046 and
0.092, 0.047 and 0.088, 0.009 and 0.066, and 0.018 and 0.044 [t hm² hr/(MJ mm hm²)] for $K_{EPIC}$, $K_{NOMO}$,
$K_{M-NOMO}$, $K_{Torri}$, and $K_{Shirazi}$ respectively. The range of the maximum values were 1.875, 2.000, 1.872, 7.333 and
2.444 times larger than the corresponding minimum values (Table 2). The differences between the mean and




median values were 0.001, -0.001, 0.000, 0.000, and 0.000 [t $hm^2$ hr/(MJ mm $hm^2$)], respectively. The standard
deviations (SDs) of the $K$ values were 0.408, -0.447, -1.079, -2.639, and 0.059, respectively, and the skewnesses
of the $K$ values were 0.946, 0.956, 4.353, 16.872, and 0.009, respectively. The $Cv$ value of $K_{M\text{-}NOMO}$ was 0.067
＜ 10 %; in addition, the $Cv$ values of $K_{EPIC}$, $K_{NOMO}$, $K_{Torri}$, and $K_{Shirazi}$ were 0.109, 0.110, 0.113, and 0.182,
respectively, all of which were between 10 % and 100 %.

In the Taylor diagrams (Taylor, 2001) (Fig. 3), the $K$ values based on EPIC model is used as the reference

object. The $K$ values based on Torri, NOMO, and Shirazi models were similar and were located close to each
other. In contrast, there was inconsistency in the $K$ values estimated by M-NOMO and EPIC models.

**3.2 Spearman correlation coefficients between soil erodibility and environmental variables in Ansai**

**watershed**

The correlations between soil erodibility and the environmental variables varied with the different

vegetation types (Table S1-S4). In general, soil erodibility in artificially managed vegetation types (apple
orchards and David's peach) and artificially restored vegetation types (e.g., sea buckthorn and black locust) had
significant correlation with vegetation properties. For example, soil erodibility in areas planted with apple
orchards had a significant positive correlation with plant density (Table S1). The soil erodibility of areas with sea
buckthorn had significant negative correlations with the slope gradient and plant density, whereas it had
significant positive correlations with the average annual rainfall and aboveground biomass (Table S3). The soil
erodibility of areas with David's peach had a significant positive correlation with the aboveground biomass,
whereas it had significant negative correlations with the slope gradient, vegetation coverage, vegetation height,
crown width and basal diameter (Table S4). The soil erodibility of areas with black locust had a significant
negative correlation with the elevation, whereas it had significant positive correlations with the slope position,
slope gradient, soil bulk density, vegetation coverage, litter biomass and branch number (Table S4). Meanwhile,





soil erodibility in areas under different vegetation types such as grasslands or farmlands were more correlated
with soil or landscape properties. The results of the correlation analysis between estimated $K$ values and the
selected environmental variables showed that soil erodibility in farmlands had significant positive correlations
with the slope position, slope shape and average annual rainfall and displayed a significant negative correlation
with the slope gradient (Table S1). Soil erodibility of areas with native grasslands had a significant negative
correlation with the elevation, whereas it had significant positive correlations with the average annual rainfall
and slope gradient (Table S2). Soil erodibility of areas with pasture grasslands did not have significant
correlations with the environmental variables other than soil organic matter content and the soil particle size
(Table S2). The soil erodibility of areas with *Caragana korshinskii* had a significant positive correlation with the
elevation, whereas it had a significant negative correlation with the average annual rainfall (Table S3).
**3.3 Principal component analysis of soil erodibility under different vegetation types**
Our results showed the PCA identified one PC each for apple orchards, native grasslands, sea buckthorn,
*Caragana korshinskii* and pasture grasslands, which accounted for 100%, 48.88%, 62.05% and 53.61 of the
variances, respectively (Table S5). The PCA identified two PCs each for farmland and David's peach; the
corresponding cumulative variances were 73.93 % and 81.07 %, respectively. For black locust, the PCA
identified three PCs that accounted for 70.25 % of the variance (Table S5). In farmland, PC1 included two
variables that had highly weighted factor loadings, the slope shape and slope position, and PC2 included only the
slope gradient, which had a highly weighted factor loading. In apple orchards, the highly weighted factor loading
was the plant density. In native grasslands, PC1 included two variables that had highly weighted factor loadings,
including the slope gradient and elevation. The pasture grasslands had no variables with highly weighted factor
loadings because it had no significant environmental variables except the soil particle size and soil organic
matter. The highly weighted factor loadings in areas with sea buckthorn were the slope gradient, aboveground





biomass and plant density. In areas planted with *Caragana korshinskii*, two variables had highly weighted factor
loadings, including the average annual rainfall and elevation. In areas planted with black locust, the highly
weighted factor loadings of PC1 were the slope position, elevation and litter biomass; for PC2, the slope gradient
and soil bulk density had high factor loadings, whereas only vegetation coverage had a high weighted factor
loading for PC3. In areas planted with David's peach, PC1 included three variables that had highly weighted
factor loadings, specifically the crown width, vegetation height and vegetation coverage, whereas only the basal
diameter had a high factor loading for PC2 (Table S5).

The MDS of the soil erodibility included six environmental variables for black locust, four for David's

peach, three each for farmland and sea buckthorn, two each for native grasslands and *Caragana korshinskii*, one
for apple orchards and none for pasture grasslands (Table 3). In addition to the soil organic matter and soil
particle size, which are included in the $K$ value estimation equations, the dominant factors affecting the soil
erodibility for farmland were slope shape, slope gradient and slope position. For apple orchards, the only
dominant factor affecting soil erodibility (except the soil organic matter and soil particle size) was plant density.
For areas with native grasslands, the dominant factors affecting soil erodibility were soil organic matter, soil
particle size, slope gradient and elevation. For areas with sea buckthorn, the dominant factors affecting soil
erodibility were aboveground biomass, slope gradient and plant density in addition to the two soil properties.
The dominant factors affecting soil erodibility in areas with *Caragana korshinskii* were soil particle size, soil
organic matter, average annual rainfall and elevation. For areas with black locust, the dominant factors were the
slope gradient, slope position, elevation, litter biomass, soil bulk density and vegetation coverage in addition to
the soil organic matter and soil particle size. The dominant factors affecting soil erodibility in areas with David's
peach included the soil organic matter, soil particle size, crown width, vegetation height and vegetation coverage.
**4 Discussion**





### 4.1 The optimal methods for estimating *K* values in Ansai watershed


236  In this study, we found that different models resulted in different estimations of soil erodibility (Table 2).

237 Since different estimation methods use different soil attributes as input parameters; even if the input parameters

238 are the same, the decision coefficients of the same input parameters are different. For example, the EPIC model

239 focuses on the features of the soil particle and soil nutrients, while the NOMO model focuses on not only the soil

240 particle size and soil nutrient characteristics, but also the soil structure characteristics, such as soil structure code

241 and soil permeability code. The existing soil erodibility estimation equations are used to calculate soil erodibility

242 based on data on the physicochemical soil properties, such as soil texture, soil structure, soil permeability and

243 soil organic matter content (Wischmeier et al., 1971, 1978; Williams et al., 1990; Torri et al., 1997; Shirazi et al.,

244 1988). Among these factors, the main physical soil property is the soil particle composition, such as the contents

245 of sand, silt and clay, and the main chemical soil property is the soil organic matter content (Wei et al., 2017).

246  Our results showed that the *K* values based on Torri, NOMO, and Shirazi models were are located close to

247 each other in the Taylor diagrams (Fig.3) and those three models could therefore represent the soil erodibility in

248 Ansai watershed. Based on previous studies, these models have also been recommended as the optimal models in

249 Chinese subtropical zone, purple hilly region, Northeast China, and Chinese Loss Plateau (Table 4). We, however,

250 suggested Torri and Shirazi models as better representatives of the models, based on their estimated K values and

251 the actual (measured) soil erodibility data in Ansai watershed (Zhang et al., 2001; Table S6). The estimated *K*

252 value based on Torri and Shirazi models were closer to the measured soil erodibility data among the three

253 possible appropriate models (Table 2 and Table S6). Our suggestions were also supported by a study by Lin et al.

254 (2017) who showed that the estimated *K* value based on Torri and Shirazi models was closer to the measured

255 value.

### 4.2 Environmental factors that influenced the soil erodibility


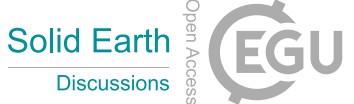

Based on the definition of K factor by Wischmeier et al. (1971), soil erodiblity is estimated by texture data,
organic matter content, soil structure index, soil permeability index. While soil erodiblity does not directly
depend on environmental factors, soil properties such as soil particle and soil organic matter can be affected by
environmental factors. Soil erodibility thus has indirect relationship with the environmental factors, particularly
vegetation type that influences the generation of soil organic matter and the composition of soil particle. Soil
erodibility had different correlation with selected environmental variables, which resulted in changes in the
dominant factors that influenced the soil erodibility (Tables S1-S5, Table 3). In native grasslands, soil erodibility
had significant correlations with terrain factors (e.g., elevation, slope degree) (Table S1, Table S4), and the
dominant factors influencing the soil erodibility were soil properties and topography. With the increase of
elevation and slope, the physical and chemical soil properties (e.g., soil permeability, soil bulk density, and soil
nutrient) and soil surface conditions are changed, further lead to the changes of soil particle size composition and
soil erodibility (Zhao et al., 2015). For example, Li et al. (2011) found that the silt content was higher than sand
in low than high elevations and Liu et al. (2005) found that slope gradient is negatively correlated with soil
nutrients (e.g., soil organic matter, available nitrogen).
For most artificially managed vegetation types (apple orchards and David's peach) and artificially restored
vegetation types (e.g., sea buckthorn and black locust), soil erodibility had significant correlations with the
vegetation properties (Table S1, Table S3-S4). By changing the physicochemical soil properties and soil structure
stability, vegetation properties could affect soil erodibility. For example, the dominant factor(s) influencing the
soil erodibility associated with apple orchards was plant density, sea buckthorn was aboveground biomass, black
locust were litter biomass and vegetation coverage, and David's peach were crown width, vegetation height,
basal diameter and vegetation coverage (Table S1). Because all these vegetation types are more or less affected
by human activities, soil erodibility can also indirectly be affected by vegetation recovery and land cover change.





## 5 Conclusions

We evaluated soil erodibility using five estimation models in Ansai watershed; the estimated $K$ values based on different models were different from one another and the resulting $K$ values ranged between 0.009 and 0.092 t hm$^2$ hr/(MJ mm hm$^2$). Based on Taylor diagrams and previous studies, we considered Shirazi and Torri model the optimal models for Ansai watershed. Since soil erodibility is estimated by soil properties, soil erodibility has indirect relationship with environment factors, including elevation and slope degree, and to a lesser extent, human activities. By changing vegetation density, biomass, and cover, human can indirectly affect soil erodibility.

**Acknowledgments** This work was supported by the National Key Research Program of China (No. 2016YFC0501604), the National Natural Science Foundation of China (No.41771197), and the State Key Laboratory of Earth Surface Processes and Resource Ecology (No. 2017-FX-01(2)). We would like to thank Jing Wang, Xiao Zhang, Qiang Feng, Xuening Fang, Jingyi Ding, and Yuanxin Liu for their support and contributions during the fieldwork.

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





**Table 1** Landscape and soil characteristics in the study area

| Vegetation types | Natural vegetation | Artificially managed vegetation | | Artificially restored vegetation | | | | |
|---|---|---|---|---|---|---|---|---|
| | NG | FL | AO | PG | SB | CK | BL | DP |
| Sampling number | 25 | 22 | 10 | 11 | 15 | 18 | 38 | 12 |
| Ele (m) | 1392.60 | 1380.14 | 1370.10 | 1401.00 | 1435.67 | 1350.61 | 1326.54 | 1377.58 |
| SG (°) | 16.72 | 6.27 | 19.90 | 11.91 | 16.40 | 17.56 | 27.24 | 24.17 |
| Cla (%) | 7.44 | 7.93 | 7.05 | 7.88 | 6.70 | 7.21 | 8.30 | 8.34 |
| Sil (%) | 45.08 | 52.63 | 48.57 | 42.73 | 45.05 | 48.08 | 51.75 | 49.69 |
| San (%) | 47.48 | 39.44 | 44.38 | 49.39 | 48.25 | 44.71 | 39.95 | 41.97 |
| OM ( g/kg) | 7.04 | 5.31 | 5.75 | 6.30 | 8.91 | 13.30 | 8.10 | 5.99 |
| SBD (g/cm$^3$) | 1.26 | 1.29 | 1.25 | 1.28 | 1.23 | 1.26 | 1.23 | 1.26 |
| Por (%) | 0.48 | 0.46 | 0.48 | 0.47 | 0.48 | 0.49 | 0.49 | 0.49 |
| AAR (mm) | 473.99 | 479.01 | 479.85 | 471.75 | 476.44 | 474.66 | 474.43 | 472.58 |
| VC (%) | 57.36 | 53.14 | 39.70 | 67.82 | 66.07 | 46.28 | 59.58 | 33.75 |
| AB (g/m$^2$) | 28.96 | 95.61 | 12.24 | 73.56 | 28.59 | 45.63 | 23.92 | 16.20 |
| VH (m) | 0.59 | 1.83 | 3.58 | 0.67 | 2.16 | 1.81 | 11.49 | 3.02 |
| LB (g/m$^2$) | 15.70 | — | 8.64 | 12.06 | 25.10 | 34.05 | 72.50 | 14.44 |
| PD (/m$^2$) | — | — | 30.50 | — | 262.40 | 131.89 | 58.66 | 36.17 |
| Cro (cm) | — | — | 398.39 | — | 184.85 | 205.20 | 448.72 | 293.40 |
| BD (cm) | — | — | 6.32 | — | 3.76 | 1.59 | 10.16 | 4.98 |
| BN | — | — | 10.17 | — | — | 27.88 | 12.86 | 8.13 |

Annotation: NG refers to native grassland, AO refers to apple orchard, FL refers to farmland, PG refers to pasture grassland, SB refers to sea
buckthorn, CK refers to *Caragana korshinskii*, DP refers to David peach, BL refers to black locust, Ele refers to elevation, SP refers to slope position,
SA refers to slope aspect, SG refers to slope gradient, SS refers to slope shape, Cla refers to clay, Sil refers to silt, San refers to sand, OM refers to
organic matter, SBD refers to soil bulk density, Por refers to porosity, AAR refers to average annual rainfall, VC refers to vegetation coverage, AB
refers to aboveground biomass, VH refers to vegetation height, LB refers to litter biomass, PD refers to plant density, Cro refers to crown, BD refers to
basal diameter, BN refers to branch number.





**Table 2** Statistics of soil erodibility in the Ansai watershed

| Methods | Samples | Mean | Max | Min | Median | SD | Skew | Kurt | *Cv* | P |
|---------|---------|------|-----|-----|--------|-----|------|------|------|---|
| EPIC | | 0.046 | 0.060 | 0.032 | 0.045 | 0.005 | 0.408 | 0.946 | 0.109 | 1.102 |
| NOMO | | 0.073 | 0.092 | 0.046 | 0.074 | 0.008 | -0.447 | 0.956 | 0.110 | 0.775 |
| M-NOMO | 151 | 0.075 | 0.088 | 0.047 | 0.075 | 0.005 | -1.079 | 4.353 | 0.067 | 0.910 |
| Torri | | 0.053 | 0.066 | 0.009 | 0.053 | 0.006 | -2.639 | 16.872 | 0.113 | 1.871 |
| Shirazi | | 0.033 | 0.044 | 0.018 | 0.033 | 0.006 | 0.059 | 0.009 | 0.182 | 1.017 |

Annotation: EPIC refers to the erosion-productivity impact model, NOMO refers to the nomograph equation, M-NOMO refers to the modified
nomograph equation, Torri refers to the $K$ value estimation model established by Torri, Shirazi refers to the $K$ value estimation model established by
Shirazi, SD refers to the standard deviation, Skew refers to the Skewness, Kurt refers to the kurtosis, $Cv$ refers to the coefficient of variation, and P
referes to p-value of Kolmogorov-Smirnov test.




**Table 3** Principal component analysis (PCA) of environmental attributes

| Vegetation types | Main influencing factors |
|---|---|
| Farmland | SS, SP, SG |
| Apple orchard | PD |
| Native grasses | SG, Ele |
| Pasture grasses | — |
| Sea buckthorn | AB, SG, PD |
| *Caragana korshinskii* | AAR, Ele |
| Black locust | SG, SP, Ele, LB, SBD, VC |
| David peach | Cro, VH, BD, VC |

Annotation: SS refers to slope shape, SP refers to slope position, SG refers to slope gradient, PD refers to plant density, Ele refers to elevation, AB
refers to aboveground biomass, AAR refers to average annual rainfall, LB refers to litter biomass, SBD refers to soil bulk density, VC refers to
vegetation coverage, Cro refers to crown, VH refers to vegetation height, BD refers to basal diameter.





**Table 4** Suggested soil erodibility estimation models in China

| Study area | optimal models | References |
|---|---|---|
| Hilly area of Chinese subtropical zone | Torri | Zhang et al.,2009 |
| Purple hilly region in Sichuan Basin | EPIC and NOMO, | Shi et al.,2012 |
| typical black soil region in Northeast China | EPIC and NOMO, | Wang et al.,2012 |
| Hilly and gully area of Chinese Loss Plateau | Torri and Shirazi | Lin et al., 2017 |
| Hilly and gully area of Chinese Loss Plateau | Shirazi | Wei et al., 2017 |



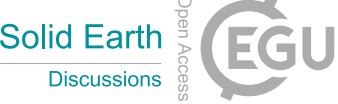

**Fig. 1** Location of the study area and the sampling points
**Fig. 2** Frequency distributions of soil erodibility
**Fig. 3** Taylor diagram were used to compare the estimating K values





Figure 1

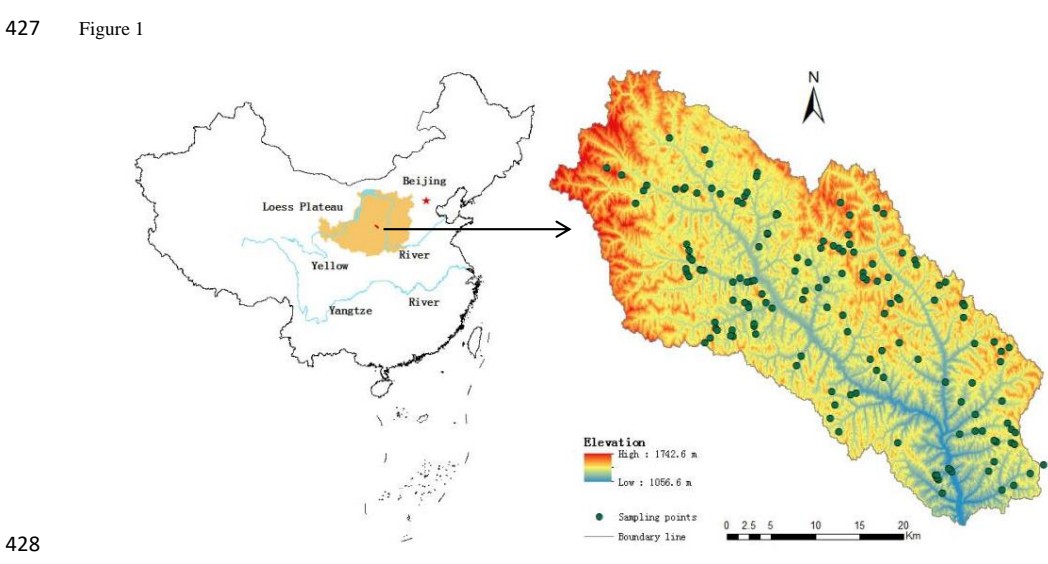





430        Figure 2

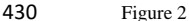

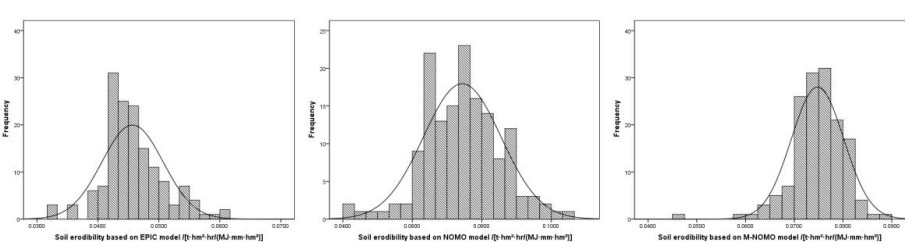


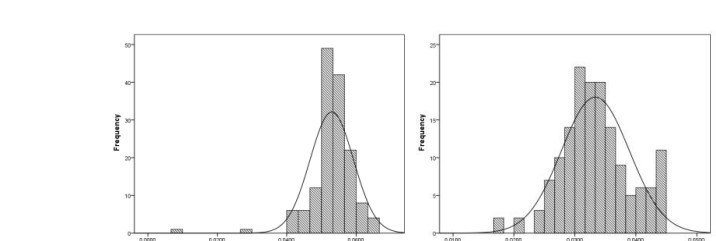







Figure 3

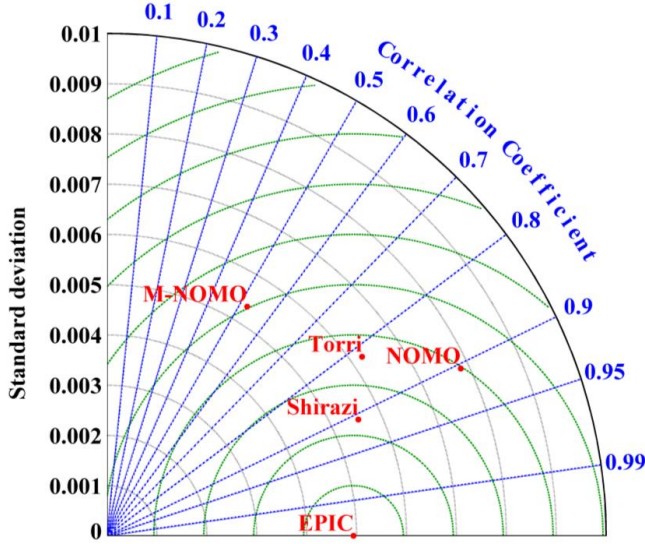
