# Peer review of "Manuscript under review for journal Solid Earth"

_Solid Earth, 2018_

## Referee Comment (RC1) · Anonymous Referee #1 · 11 Jun 2018

The manuscript presents an interesting comparison of five methods of estimating K value in a typical loess watershed. The research results that Shirazi model and Torri model are the suitable models to calculate K value will be helpful for soil erosion evaluation at local scale. I'd like to suggest the manuscript can be accepted after some revisions, and it is better to polish the language further.

The detailed suggestions are as follow: Line 30-31, Please delete the detail soil properties in bracket "(e.g., soil texture, permeability and structural stability)". Line 34 Change "research" to "researches" Line 40 Abbreviations should be added in the following of "the nomogram model and the modified nomogram model" Line 86-87 references about

the classification of soil particles should be added. Line 91-92 what is the meaning of "Soil erodibility thus has indirect relationship with the environmental factors." ? Please rewrite it. Line 106 "rainfall erosion" or "rainfall erosivity"? Line 154 Please change "P value >0.05" into "P > 0.05". Part 3.2 Significant negative or positive correlations are in P < 0.05 or P < 0.01? Need be labelled in the following. Line 202-203 One PC each for apple orchards, native grasslands, sea buckthorn, Caragana korshinskii and pasture grasslands. Why there is only 4 data of percentage in the following sentence. Please check it. Line 264 Soil erodibility has significant correlations with elevation? Please check it. If so, explain why. Line 267 What "soil surface conditions" refer to? Please give some examples. Line 277-278 What is the meaning of "Because all these vegetation types are more or less affected by human activities, soil erodibility can also indirectly be affected by vegetation recovery and land cover change."? Please rewrite it to make it easy to be understand.

---

## Referee Comment (RC2) · Anonymous Referee #2 · 30 Jun 2018

Soil erodibility is significant for the quantitative estimation of soil erosion. The manuscript, entitled Soil erodibility estimation by using five methods of estimating K value: A case study in Ansai watershed of Loess Plateau, China, tries to find the possible indirect environmental factors of soil erodibility. The topic is interesting. Nevertheless, a major revision is needed before the paper is accepted for publication.

Some of the problems in the manuscript are shown as follows: (1) Abstract. Hardly any quantitative result is found in the abstract. (2) Results. ïĆš Sections 3.2 and 3.3: where is Tables S1–S5? I try to find the relationships of the text with Tables 1-4, but I am failed. I am sure some tables have been lost in the manuscript. ïĆš Lines

220-222, Page 11: Table 3 in page 22 presents the Principal component analysis (PCA) of environmental attributes, instead of the MDS of the soil erodibility. ïĆš Some of the tables have been published in a Chinese journal. For example, Figure 2 in the manuscript is similar to Figure 3 in Reference (Zhao et al., 2017). I have uploaded the published paper together with the comments. ïĆš Too many abbreviations have been found in the manuscript. I think you may make a list for the abbreviations as an accessory of the paper. Moreover, some of the abbreviations are not needed, e.g., the words skewness and kurtosis in Table 2. ïĆš Errors exist in the annotations. Some of the annotations followed with the tables are duplicated, e.g., the annotations in Tables 1 and 3. I suggest the parameters s in the may be emerge according to their order in the table. I am sorry I could not find SP and SS in Table 1, although the terms have been explained in the annotation. ïĆš English writing of the manuscript is readable. Nevertheless many language errors exist. I strongly suggest you ask a soil scientist whose native language is English to polish the whole manuscript.

Please also note the supplement to this comment:
https://www.solid-earth-discuss.net/se-2018-43/se-2018-43-RC2-supplement.pdf

**Supplement:**

**K**

———

1 2      1 2†

(1.                           100875    ;
2.                            100875    )

:   EPIC  、  、    、Torri   Shirazi  5       ( *K* )
、  、                GIS    CSLE         2006—2014
                *K*     。   :  5
2006—2014           65. 59、106. 00、108. 47、76. 69  47. 68 t/hm² 。
      17. 73 t/hm²     ( MAE)、    ( MRE)   ( RMSE)
0   ( $A_f$ )   1              Shirazi          MAE、MRE
RMSE   30. 93、3. 25  43. 66 $A_f$  4. 41; EPIC    $A_f$   5. 80; Torri            ;
        $A_f$   7. 99  7. 88 。        4
  Shirazi                *K*
        Shirazi  。
:    *K* ;        ;    ;
: S157. 1       : A      : 2096-2673( 2017) 06-0052-14

**DOI**: 10. 16843 /j. sswc. 2017. 06. 007

**The optimal estimation method for *K* value of soil erodibility:**

**A case study in Ansai Watershed**

WEI Hui[1 2]  ZHAO Wenwu[1 2]

( 1. State Key Laboratory of Earth Surface Processes and Resources Ecology Faculty of Geographical Science
Beijing Normal University 100875 Beijing China; 2. Institute of Land Surface System and Sustainable
Development Faculty of Geographical Science Beijing Normal University 100875 Beijing China)

**Abstract**: **Background** Soil erosion has become a global ecological and environmental problem. It is now being recognized as a severe threat to socio-ecological security and stability and it is relative with the food security resilience to climate change and geosocial stability. Soil erosion is particularly acute in the Loess Plateau. In order to control soil erosion the quantitative study of soil erosion must be strengthened. Soil erosion is affected by many factors such as climate vegetation and land use and soil properties. Among those factors soil erodibility has been qualitatively evaluated as a key indicator for estimating soil loss and usually being measured by *K* value. The research of soil erodibility is significant to understand the principle of soil erosion to estimate soil erosion modulus quantitatively and to control

: 2017-05-25      : 2017-11-20
:            ”                ”(2016YFC0501604)
:   (1993—)         。  :       。 E-mail: irene1993weihui@ 163. com
†   :  (1976—)          。  :       。 E-mail: zhaoww@ bnu. edu.
   cn

soil and water loss reasonable. The estimation method of soil erodibility is numerous  but the regional applicability of different models remains to be discussed.  **Methods**  We conducted a study to select the optimal estimation method of soil erodibility ( *K* value)  based on the basic data of precipitation  soil remote sensing images and socioeconomic data in Ansai Watershed. We used GIS technology and CSLE model to calculate soil erosion modulus in 2006 − 2014 and compared it with the corresponding monitoring value of sediment. Tthe *K* value was calculated by EPIC  NOMO  M‒NOMO  Torri and Shirazi model. The comparison between the simulated value of soil erosion modulus and the monitoring value of sediments is based on the principle that the mean absolute error ( MAE)  the mean relative error ( MRE)  and the root‒mean‒square error ( RMSE)  are closer to 0  the accuracy factor ( $A_f$ )  is closer to 1  the regional applicability of model is higher.  **Results**  The mean soil erosion modulus of Ansai watershed in 2006‒ 2014 based on the five models of EPIC  NOMO  MNOMO  Torri  and Shirazi was 65. 59  106. 00 108. 47  76. 69 and 47. 68 t/hm$^2$ respectively. The mean monitoring value of sediment in corresponding year was 17. 73 t/hm$^2$. Based on the above evaluation indexes  we knew that the Shirazi model′s regional applicability was the highest  the value of MAE  MRE and RMSE was 30. 93  3. 25 and 43. 66 respectively  the $A_f$ value was 4. 41  The regional applicability of EPIC model took second place  the value of $A_f$ was 5. 80  The regional applicability of Torri model was in the middle level. The regional applicability of NOMO model and M‒NOMO model was lowest  it had the biggest difference with the actual situation  the $A_f$ value was 7. 99 and 7. 88 respectively.  **Conculsions**  Based on the above analysis  we concluded that the Shirazi model had the best applicability in study area comparing to the other four *K* value estimation methods. We should be preferred to choose the Shirazi model in the future watershed scale soil erodibility ( *K* value)  estimation and soil erosion evaluation.

**Keywords**: soil erodibility ( *K* value) ;  China soil loss equation;  model optimization;  Ansai Watershed

1 334 km²

5                                          505.3 mm    74%          6—9

*K*                  CSLE              2006—        8.8 ℃    >10 ℃    2 876~3 270 ℃

2014                                        2 397.3 h。

*K*

997~

1 731 m

**1**

(E 108°5´44″–109°26´18″ N 36°    97%

30´45″–37°19´3″)                                        29

**Fig.1**  Location of the study area

30 m×30 m                            )

**2**

**2.1                              2 m×2 m                5 m×5 m**

:                    10 m×10 m          GPS

25 m×25 m                (DEM)

1:50 000          ; 2005    **2.2**

2015    2                                        (China Soil Loss Equation)

;                《          》        USLE          RUSLE

》;

(          );        2006—2014                      30

;    、                              $A = R \cdot K \cdot L \cdot S \cdot B \cdot E \cdot T$。          (1)

《          》;      :*A*                t/(hm²·a);*R*

2014    7—8                    MJ·mm/(hm²·h·a);*K*

151                          t·hm²·h/(hm²·MJ·mm);*L*            ;*S*

8              ;*B*              ;*E*

(    、        、    、        ;*T*                。

、    、    、    )。        2.2.1                          *K*

>2 km            (        16(NOMO)、        17(M–NOMO)、EPIC

[Figure]

**Fig. 2** Distribution of sampling points the study area

[18]、Torri [19] Shirazi [20] 5

*K* [16-20]。 *K*

**2.2.2** *R*

[31]

[32]

**2.2.3** *LS*

*LS*

DEM ;

DEM

*LS* [33-34]

*LS*

25 m DEM

[35]

[36]

[33]

90%

[33]

**2.2.4** *B*

[37] 0 ~ 1

NDVI

*B*

*B* [38]

**2.2.5** *E*

[36]

[38]

*E*

**2.2.6** *T*

[31 39] 0 ~ 1 T

*T* [38]

**3.1** CSLE

**3.1.1** *K*

1) 。 *K*

*K*

( 1)。

**1** *K*

**Tab. 1** Statistics characteristics of soil erodibility factor value ( *K* )

| Method | Samples | Mean | Max | Min | Median | ± SD | Skew | Kurt | $C_v$ | *K*-SP |
|---|---|---|---|---|---|---|---|---|---|---|
| EPIC | | 0. 046 | 0. 060 | 0. 032 | 0. 045 | 0. 005 | 0. 408 | 0. 946 | 0. 109 | 1. 102 |
| NOMO | | 0. 073 | 0. 092 | 0. 046 | 0. 074 | 0. 008 | − 0. 447 | 0. 956 | 0. 110 | 0. 775 |
| M–NOMO | 151 | 0. 075 | 0. 088 | 0. 047 | 0. 075 | 0. 005 | − 1. 079 | 4. 353 | 0. 067 | 0. 910 |
| Torri | | 0. 053 | 0. 066 | 0. 009 | 0. 053 | 0. 006 | − 2. 639 | 16. 872 | 0. 113 | 1. 871 |
| Shirazi | | 0. 033 | 0. 044 | 0. 018 | 0. 033 | 0. 006 | 0. 059 | 0. 009 | 0. 182 | 1. 017 |

1 $K_{EPIC}$、$K_{NOMO}$、$K_{M-NOMO}$、$K_{Torri}$ $K_{Shirazi}$ 0. 047 ~ 0. 088、0. 009 ~ 0. 066 0. 018 ~ 0. 044
0. 032 ~ 0. 060、0. 046 ~ 0. 092、 1. 875、2. 000、1. 872、

7.333   2.444           $K$                    。 $K$      、    、
                                    0       $K$                                                                                    、                      、
                                    $K$                                   。       、                                                        。
$K_{\text{M-NOMO}}$    $C_v$        0.067 < 10%         $K_{\text{M-NOMO}}$                                    2)                 。         SPSS 20.0         $K$
                                                       ;                   $K_{\text{EPIC}}$、                    ( 3)。        5                 $K$
$K_{\text{NOMO}}$、$K_{\text{Torri}}$     $K_{\text{Shirazi}}$   $C_v$               0.109、0.110、                                                                          40         。
0.113   0.182        10% ~ 100%                                          K–S                  K–SP        > 0.05
                                    。                                      $K$

[Figure]

**图 3**                $K$

**Fig. 3**    Frequency distribution of soil erodibility factor value ($K$)

        3)                    。                                                $K$
                                                                                          Kriging
           、                3               40    。         $K$           40    。
                                    $K$
    ;                                        $K$             Kriging        $K$                    ( 4)。
                   ;               $K$                    、            4        $K_{\text{EPIC}}$、$K_{\text{NOMO}}$、$K_{\text{Torri}}$      $K_{\text{Shirazi}}$                                          $K$
ArcGIS 10.1                                                                                                              。
        ( 2)。
        2                                    $K$                                                                                                    。
                           。 $K_{\text{EPIC}}$                                                    $K_{\text{M-NOMO}}$
           $C_0/( C_0 + C)$         11% < 25%                                            、
$K_{\text{EPIC}}$                                                     ; $K_{\text{NOMO}}$、
$K_{\text{M-NOMO}}$、$K_{\text{Torri}}$     $K_{\text{Shirazi}}$                                                                                      。
$C_0/( C_0 + C)$              52%、49%、54%     42%          3.1.2                        $R$                        20
    25% ~ 75%                                                                                    Krig–
        。       61.09 m            733.06 m                  ing         $R$                    ( 5)。

**2** K

**Tab. 2** Semivariance function analysis results of soil erodibility factor value (K)

| Method | Model | Nugget $\times 10^{-6}$ | Partial sill $\times 10^{-6}$ | Sill $\times 10^{-6}$ | $C_0/(C_0+C)$ | Lag/m | Range/m |
|---|---|---|---|---|---|---|---|
| EPIC | Spherical | 8.06 | 18.54 | 26.61 | 0.30 | 61.09 | 733.06 |
| | Exponential | 2.91 | 24.08 | 27.00 | 0.11 | 61.09 | 733.06 |
| | Gaussian | 12.78 | 14.78 | 27.55 | 0.46 | 61.09 | 733.06 |
| NOMO | Spherical | 72.89 | 53.82 | 126.71 | 0.58 | 61.09 | 733.06 |
| | Exponential | 80.32 | 42.05 | 122.37 | 0.66 | 61.09 | 733.06 |
| | Gaussian | 72.24 | 65.97 | 138.22 | 0.52 | 61.09 | 733.06 |
| M-NOMO | Spherical | 30.39 | 16.90 | 47.29 | 0.64 | 61.09 | 733.06 |
| | Exponential | 31.79 | 0 | 31.79 | 1.00 | 61.09 | 733.06 |
| | Gaussian | 29.44 | 30.89 | 60.33 | 0.49 | 61.09 | 733.06 |
| Torri | Spherical | 26.49 | 20.80 | 47.29 | 0.56 | 61.09 | 733.06 |
| | Exponential | 32.17 | 14.32 | 46.49 | 0.69 | 61.09 | 733.06 |
| | Gaussian | 26.40 | 22.29 | 48.69 | 0.54 | 61.09 | 733.06 |
| Shirazi | Spherical | 13.12 | 14.92 | 28.05 | 0.47 | 61.09 | 733.06 |
| | Exponential | 20.60 | 4.96 | 25.56 | 0.81 | 61.09 | 733.06 |
| | Gaussian | 12.78 | 17.57 | 30.35 | 0.42 | 61.09 | 733.06 |

[Figure]

**4** K

**Fig. 4** Spatial distribution of soil erodibility factor K in the study area

2006—2014 R 1 365.06、 1 811.11、4 416.59 1 765.42 MJ·mm/(hm²·h·
1 416.39、820.89、2 560.23、1 251.06、1 072.43、 a)。 5 2008 2014 R

[Figure]

**图 5     研究区降雨侵蚀力因子 R 的空间分布**

**Fig. 5**  Spatial distribution of rainfall erosivity factor $R$ in the study area

。      2009    2012              $R$

                              ; 2011    2013    $R$      95%       $LS$        27

                                    ;                    $LS$

2008    2014    $R$

              ;        $R$

                                                    ArcGIS 10. 1              $B$

    。                                          (   7)。

3. 1. 3              $LS$                                7      $B$

                                                                              。   2006—2011    $B$

    $LS$(   6)。

    6      $LS$              0. 02                    $B$

78. 12          11. 76。              $LS$        5

                                                                    20% ; $LS$

                                                                    60% ;

                                                                                              [38]

                                                    3. 1. 4              $B$

                                                                              $B$

                                                                              $B$

                                                                              ; 2012    2013

                                                                              ; 2014

                                                                              $B$

[Figure]

**6** LS

**Fig. 6** Spatial distribution of slope steepness factor *LS* in study area

B 。 B 2006 2014 B 0. 11 0. 15。 ( ) B 。 2006 139. 14 km² 2014 115. 18 km² 975. 80 km² 1 041. 53 km²。

3. 1. 5 E 、 2006—2014 E ( 3)。

3 E 0. 84 ~ 0. 88 。 2011 E 0. 84 2013 2014 E 0. 88。

2006
0.004~0.020
0.020~0.060
0.060~0.090
0.090~0.450
0.450~1.000

2007
0.004~0.020
0.020~0.060
0.060~0.090
0.090~0.450
0.450~1.000

2008
0.004~0.020
0.020~0.060
0.060~0.090
0.090~0.450
0.450~1.000

2009
0~0.004
0.004~0.039
0.039~0.086
0.086~0.447
0.447~1.000

2010
0~0.004
0.004~0.039
0.039~0.086
0.086~0.447
0.447~1.000

2011
0.004~0.020
0.020~0.059
0.059~0.149
0.149~0.238
0.238~1.000

2012
0.004~0.020
0.020~0.059
0.059~0.149
0.149~0.450
0.449~1.000

2013
0.004~0.039
0.039~0.059
0.059~0.098
0.098~0.449
0.449~1.000

2014
0.004~0.059
0.059~0.098
0.098~0.149
0.149~0.449
0.449~1.000

**7** B

**Fig. 7** Spatial distribution of *B*-factor in study area

**3**                                                                                                                        (*E*)
**Tab. 3**   Engineering measure factor value in the study period (*E*)

| Year | 2006 | 2007 | 2008 | 2009 | 2010 | 2011 | 2012 | 2013 | 2014 |
|---|---|---|---|---|---|---|---|---|---|
| *E*   *E* value | 0.85 | 0.85 | 0.85 | 0.85 | 0.85 | 0.84 | 0.85 | 0.88 | 0.88 |

3.1.6                      *T*              DEM                          。          2013

                                                    *T*                                                              、

            (    8)。                                                                           2014

[Figure]

                                                                                          。

                                                                               **3.3**                 *K*

                                                                                 ArcGIS 10.1                  2006—2014

                                                                                              (MAE)、                (MRE)、

                                                                    (RMSE)              (*A*$_f$)

                                                                          5      *K*                                          。

                                                    MAE、MRE    RMSE              0 *A*$_f$              1    *K*

                                                                                                                       [23−42]。

0.10
0.10~0.22
0.22~0.30
0.30~0.57
0.57~0.80
0 5 10 15 20 km

                                                                                                            24

            **8**          *T*                                  4        。

**Fig. 8**   Spatial distribution of tillage factor (*T*)                     5                5    *K*

            in the study area                                                                          。

        8                  *T*                    0.10                    65.59、106.00、108.47、76.69

0.80              0.64。   *T*                47.68 t/hm²。

                                *T*                17.73 t/hm²    MAE、MRE    RMSE

    *T*                。                                  0 *A*$_f$          1

**3.2**                  *K*                                    Shirazi                    MAE、MRE    RMSE

            5            *K*    CSLE                30.93、3.25    43.66 *A*$_f$        4.41 EPIC

                  ArcGIS 10.1                            *A*$_f$        5.80 Torri

                                。                                                *A*$_f$          7.99   7.88。

            SL190—2007 [41]                                    Shirazi        EPIC

            (≤500)、    (500~2 500)、                                      *K*

(2 500~5 000)、    (5 000~8 000)、                。

(8 000~15 000)    (>15 000)6                EPIC            *K*  [12 21 38 43]

                        (    9)。                *K*

        9                *K*                    Shirazi                *K*              。

                        。      :                    [23]

2006                                                                            EPIC

            ; 2008                                [24]                          *K*          ;

                ; 2011                                              Torri        *K*

                    。                                                          。

2009、2013    2014                                                                      *K*

        2013                              。                  *K*

                2009    2013                                      *K*                            。

2006(a) 2006(b) 2006(c) 2006(d) 2006(e)

2007(a) 2007(b) 2007(c) 2007(d) 2007(e)

2008(a) 2008(b) 2008(c) 2008(d) 2008(e)

2009(a) 2009(b) 2009(c) 2009(d) 2009(e)

2010(a) 2010(b) 2010(c) 2010(d) 2010(e)

2011(a) 2011(b) 2011(c) 2011(d) 2011(e)

2012(a) 2012(b) 2012(c) 2012(d) 2012(e)

**Fig. 9** Spatial distribution of soil erosion in the study area

[Figure]

a b c d e          EPIC、        、         、Torri      Shirazi

a b c d e refer to the results based on EPIC  NOMO  M-NOMO  Torri  Shirazi model respectively.

9( )

**Fig. 9(Continued)**    Spatial distribution of soil erosion in the study area

**4**                   $K$

**Tab. 4**    Regional suitability evaluation of different $K$ value models in watershed area

| Year | Monitoring value of sediment/(t·hm$^{-2}$) | Simulation value of soil erosion modulus/(t·hm$^{-2}$) | | | | |
|---|---|---|---|---|---|---|
| | | EPIC | NOMO | M-NOMO | Torri | Shirazi |
| 2006 | 19. 90 | 44. 47 | 71. 83 | 73. 60 | 51. 97 | 32. 32 |
| 2007 | 37. 26 | 39. 11 | 63. 12 | 64. 71 | 45. 73 | 28. 42 |
| 2008 | 9. 00 | 24. 13 | 38. 96 | 39. 90 | 28. 20 | 17. 54 |
| 2009 | 12. 10 | 106. 29 | 171. 53 | 176. 38 | 124. 59 | 77. 07 |
| 2010 | 18. 80 | 39. 85 | 64. 34 | 65. 99 | 46. 63 | 28. 94 |
| 2011 | 6. 84 | 32. 40 | 52. 35 | 53. 72 | 37. 96 | 23. 56 |
| 2012 | 11. 70 | 33. 91 | 54. 72 | 56. 19 | 39. 72 | 24. 61 |
| 2013 | 40. 00 | 179. 74 | 290. 90 | 296. 68 | 209. 87 | 130. 86 |
| 2014 | 3. 94 | 90. 37 | 146. 24 | 149. 08 | 105. 52 | 65. 79 |
| Average | 17. 73 | 65. 59 | 106. 00 | 108. 47 | 76. 69 | 47. 68 |
| MAE | 47. 86 | 88. 27 | 90. 75 | 58. 96 | 30. 93 | |
| MRE | 4. 77 | 8. 33 | 8. 54 | 5. 75 | 3. 25 | |
| RMSE | 65. 23 | 114. 93 | 117. 82 | 78. 76 | 43. 66 | |
| $A_f$ | 5. 80 | 7. 88 | 7. 99 | 6. 48 | 4. 41 | |

1)      5                $K$

$K_{\text{M-NOMO}} > K_{\text{NOMO}} > K_{\text{Torri}} > K_{\text{EPIC}} > K_{\text{Shirazi}}$。

$K_{\text{EPIC}}$、$K_{\text{NOMO}}$、$K_{\text{Torri}}$    $K_{\text{Shirazi}}$

              K

                      。

。                              $K_{\text{M-NOMO}}$

`、

                                              。

2)      5       $K$

                            。      EPIC、         、         、

Torri       Shirazi         2006—2014

              65. 59、106. 00、108. 47、76. 69

47. 68 t/hm$^2$。

17. 73 t/hm² Shirazi
EPIC Torri 5

;

4 K Shirazi

K

K

Shirazi 。

1 TRIMBLE S W CROSSON P. Soil erosion rates-myth and reality J . Science 2000 289: 248.

2 ZHU Mingyong. Soil erosion assessment using USLE in the GIS environment: a case study in the Danjiangkou Reservoir Region China J . Environmental Earth Sciences 2015 73(12): 7899.

3 PIMENTEL D HARVEY C RESOSUDARMO P et al. Environmental and economic costs of soil erosion and conservation benefits J . Science 1995 267(5201): 1117.

4 MARZEN M ISERLOH T LIMA J L M P D et al. Impact of severe rain storms on soil erosion: experimental evaluation of wind-driven rain and its implications for natural hazard management J . Science of the Total Environment 2017 590/591: 502.

5 . M . : 2002: 185.
FU Bojie CHEN Liding QIU Yang et al. Land use structure and ecological processes in the Loess Hillyarea M . Beijing: The Commercial Press 2002: 185.

6 . J . 2012 27(3): 346.
SHAN Lun. Soil and water conservation and sustainable development J . Bulletin of Chinese Academy of Scinece 2012 27(3): 346.

7 . 、 J . 2011(8): 148.
LI Yonghong GAO Zhaoliang. The Loess Plateau area the characteristics of soil and water loss damages and management J . Ecological Economy 2011(8): 148.

8 Romkens M J M. J . 2013 20(1): 277.
WANG Bin ZHENG Fenli Romkens M J M. Soil erodibility for water erosion: a review J . Research of Soil and Water Conservation 2013 20(1): 277.

9 RODRIGUEZ-ITURBE I D´ODORICO P LAIO F et al. Challenges in humid land ecohydrology: interactions of water table and unsaturated zone with climate soil and vegetation J . Water Resources Research 2007 43(9): W09301.

10 WANG Bin ZHENG Fenli ROMKENS M J M et al. Soil erodibility for water erosion: a perspective and Chinese experiences J . Geomorphology 2013 187 (187): 1.

11 . J . 2007 44(1): 7.
ZHANG Keli PENG Wenying YANG Hongli. Soil erodibility and its estimation for agricultural soil inChina J . Acta Pedologica Sinica 2007 44(1): 7.

12 . J . 2013 24 (1): 105.
GAO Liqian ZHAO Yubge QIN Ningqiang et al. Effects of biological soil crust on soil erodibility in Hilly Loess Plateau region of northwest China J . Chinese Journal of Applied Ecology 2013 24(1): 105.

13 . K J . 2015(6): 1192.
CAO Xianghui LONG Huaiyu LEI Qiuliang et al. Assessment and analysis of the topsoil erodibility K values in Hebei province J . Soils 2015(6): 1192.

14 BONILLA C A JOHNSON O I. Soil erodibility mapping and its correlation with soil properties in Central Chile J . Geoderma 2012 s 189/190(2): 116.

15 . J . 2011 18(1): 77.
GU Shixian WANG Xiaodan LIU Shuzhen. The preliminary research on the model method of soil erodibility in the Aixigou Watershed of Jinsha River J . Research of Soil and Water Conservation 2011 18(1): 77.

16 WISCHMEIER W H JOHNSON C B CROSS B V. soil erodibility nomograph for farmland and construction sites J . Journal of Soil & Water Conservation 1971 26 (5): 189.

17 WISCHMEIER W H SMITH D D. Predicting rainfall erosion losses-a guide to conservation planning J . United States. Dept. of Agriculture. Agriculture Handbook 1978 537.

18 WILLIAMS J R. The Erosion-Productivity Impact Calculator (EPIC) model: a case history J . Philosophical Transactions of the Royal Society B Biological Sciences 1990 329(1255): 421.

19    TORRI D  POESENJ  BORSELLI L. Predictability and
      uncertainty of the soil erodibility factor using a global
      dataset  J . Catena  1997  31(1/2):1.

20    SHIRAZI M A  HART J W  BOERSMA L. A unifying
      quantitative analysis of soil texture: improvement of pre-
      cision and extension of scale  J . Soil Science Society
      of America Journal  1988  52(1):181.

21                                 .            /
                        J .                 2009  25(2):56.
      ZHU Bingbing  LI Zhanbin  LI Peng  et al. Dynamic
      changes of soil erodibility during process of landdegrada-
      tion and restoration  J . Transactions of the CSAE
      2009  25(1):181.

22                                 .
                             J .                 2014  22
      (4):743.
      ZENG Quanchao  LI Yaru  LIU Lei  et al. Study on
      soil aggregate stability and soil erodibility in the grass-
      land vegetation of the Loess Plateau Region  J . Acta
      Agrestia Sinica  2014  22(4):743.

23                                 .
             K                     J .
         2012  34(1):33.
      SHI Dongmei  CHEN Zhengfa  JIANG Guangyi  et al.
      Estimation methods for soil erodibility $K$ in purple area
       J . Journal of Beijing Forest University  2012  34
      (1):32.

24                                 .
             K                     J .                 2009
      46(2):185.
      ZHANG Wentai  YU Dongsheng  SHI Xuezheng  et al.
      Uncertainty in prediction of soil erodibility $K$-factor in
      subtropical China  J . Acta Pedologica Sinica  2009
      46(2):185.

25                                 .
                        J .                 2009  7(6):
      113.
      LI Linyu  JIAO Juying  CHEN Yang. Research meth-
      ods and results analysis of sediment delivery ratio  J .
      Science of Soil and Water Conservation  2009  7(6):
      113.

26                                 .                 J .
         1979(1):9.
      GONG Shiyang  XIONG Guishu. The source and distri-
      bution of sediment in the Yellow River  J . 1979(1):
      9.

27                                 .
             J .                 1982(2):62.

      MOU Jinze  MENG Qingmei. The sediment transport
      ratio of a basin in the calculation of sediment yield  J .
      Journal of Sediment Research  1982(2):62.

28                                 .            J .
      2002(1):53.
      JING Ke. Sediment delivery ratio in the upper Yangtze
      River  J . Journal of Sediment Research  2002(1):
      53.

29                                 .
                                 :
                        J .                 2014  34(5):1105.
      ZHAO Mingyue  ZHAO Wenwu  ZHONG Lina. Scale
      effect analysis of the influence of land use andenviron-
      ment factors on surface soil organic carbon: a case study
      in the hilly and gully area of Northern Shanxi Province
       J . Acta Ecologica Sinica  2014  34(5):1105.

30    LIU Baoyuan  ZHANG Keli  XIE Yun. An empirical
      soil loss equation  C //Proceedings-Process of soil ero-
      sion and its environment effect  12th international soil
      conservation organization conference  Tsinghua Universi-
      ty Press  Beijing  2002  21.

31                                 .       RUSLE
                                 :                 J .
         2015  35(3): 365.
      YI Kai  WANG Shiyang  WANG Xue  et al. The char-
      acteristics of spatial-temporal differentiation of soil ero-
      sion based on RUSLE model: a case study of Chaoyang
      city  Liaoning province  J . Scientia Geographica Sini-
      ca  2015  35(3): 365.

32                                 .
             J .                 2003  25(1):35.
      ZHANG Wenbo  FU Jinsheng. Rainfall erodibility esti-
      mation under different rainfall amount  J . Resources
      Science  2003  25(1):35.

33                                 .            DEM
                                 :
             J .                 2001  21(1):53.
      TANG Guoan  YANG Qinke  ZHANG Yong  et al. Re-
      search on accuracy of slope derived from DEMs ofdiffer-
      ent map scales  J . Bulletin of Soil and Water Conser-
      vation  2001  21(1):53.

34                                 .
                                 :                 J .
         2015  34(8):1039.
      DING Jingyi  ZHAO Wenwu  WANG Jun  et al. Scale
      effect of the impact on runoff of variations inprecipitation
      vegetation: taking northern Shanxi loess hilly-gully re-
      gion as an example  J . Progress in Geography  2015

   34(8):1039.

35                .DEM

          J.         2016

   14(5):15.

   LI Mengmeng ZHAO Yuanyuan GAO Guanglei et al.
   Effects of DEM resolution on the accuracy of topographic
   factor derived from DEM J.Science of Soil and Water
   Conservation 2016 14(5):15.

36   LIU Baoyuan NEARING M A SHI Peijun et al.
   Slope length effects on soil loss for steep slopes J.
   Soil Science Society of America Journal 2000 64(5):
   1759.

37             .

               J.

   2002 13(8):1033.

   ZHANG Yan YUAN Jianping LIU Baoyuan. Advance
   in researches on vegetation cover and management factor
   in the soil erosion prediction model J.Chinese Jour-
   nal of Applied Ecology 2002 13(8):1033.

38            .     ( )

              J.

   2009 42(2):569.

   XIE Hongxia LI Rui YANG Qinke et al. Effect of re-
   turning farmland to forest(pasture)and changes ofpre-
   cipitation on soil erosion in the Yanhe Basin J.Scien-
   tia Agricultura Sinica 2009 42(2):569.

39   XU Lifen XU Xuegong MENG Xiangwei. Risk assess-
   ment of soil erosion in different rainfall scenarios by RU-
   SLE model coupled with information diffusion model: a

case study of Bohai Rim China J.Catena 2012
100: 74.

40              .

       *K*       J.     2008 28(5):
2199.

ZHANG Jinching LI Haidong LIN Jie et al. Spatial
variability of soil erodibility(*K*-factor)at a catchment
scale in China J.Acta Ecologica Sinica 2008 28
(5):2199.

41              .          :

SL190—2007 S.   :
2007.

The Ministry of Water Resources of the People′s Repub-
lic of China. Standards for classification and gradation of
soil erosion: SL190 − 2007 S.Beijing: China Water
& Power Press 2007.

42   DONG Qingli TU Kang GUO Liyang et al. Response
surface model for prediction of growth parameters from
spores of *Clostridium sporogenes* under different experi-
mental conditions J.Food Microbiology 2007 24
(6):624.

43               .

              J.       2016
30(5):89.

ZHAO Wenqi LIU Yu LUO Mingliang et al. Effect
of re-vegetation on soil erosion in small watershed of the
Loess Plateau J.Journal of Soil and Water Conserva-
tion 2016 30(5):89.

---

## Author Comment (AC1) · 2 Aug 2018

Dear reviewer #1, We greatly appreciate the work that you have put into our manuscript and the support to our work. The comments inspired us to pay more attention to the accuracy of our language and the details of explanation in our manuscript. The following is a point-by-point response to all of the comments. We hope that the response will meet your approval. Once again, thank you very much for your comments and suggestions. Best regards,

Wenwu Zhao Email: zhaoww@bnu.edu.cn

[Figure]

Point-by-point responses: 1. It is better to polish the language further. Reply: The language of our manuscript will be polished by a native English speaker. 2. Line 30-31, Please delete the detail soil properties in bracket "(e.g., soil texture, permeability and structural stability)". Reply: The detail soil properties in bracket will be deleted. 3. Line 34 Change "research" to "researches" Reply: The "research" will be changed to "researches". 4. Line 40 Abbreviations should be added in the following of "the nomogram model and the modified nomogram model" Reply: The abbreviations will be added behind the following phrases "the nomogram model and the modified nomogram model". 5. Line 86-87 references about the classification of soil particles should be added. Reply: The relevant references on classification of soil particles will be added. 6. Line 91-92 what is the meaning of "Soil erodibility thus has indirect relationship with the environmental factors." ? Please rewrite it. Reply: The meaning of this sentence is that the environmental factors may not have direct influence on soil erodibility but they can affect soil erodibility by changing the soil particle and soil organic matter content. This sentence will be rewritten to make it easier to read. 7. Line 106 "rainfall erosion" or "rainfall erosivity"? Reply: The "rainfall erosion" will be changed to "rainfall erosivity". 8. Line 154 Please change "P value >0.05" into "P > 0.05". Reply: The "P value >0.05" will be changed to "P > 0.05". 9. Part 3.2 Significant negative or positive correlations are in P < 0.05 or P < 0.01? Need be labelled in the following. Reply: The "P < 0.05" or "P < 0.01" will be added in the following of each correlation analysis. 10. Line 202-203 One PC each for apple orchards, native grasslands, sea buckthorn, Caragana korshinskii and pasture grasslands. Why there is only 4 data of percentage in the following sentence. Please check it. Reply: The analysis of pasture grasslands did not come out a result, so there are only 4 data of percentage. The explanation of this analysis will be revised in this and the following sentence to make it more accurate and clearer. 11. Line 264 Soil erodibility has significant correlations with elevation? Please check it. If so, explain why. Reply: The soil erodibility showed significant correlations with elevation in this paper. According to existing studies, for a limited area, the elevation might have relationship with factors such as soil type, vegetation type which have significant

correlations with soil erodibility. Relevant discussion will be added in the manuscript to explain the reason of this result. 12. Line 267 What "soil surface conditions" refer to? Please give some examples. Reply: Soil surface condition is a collection of many factors such as surface roughness and vegetation coverage, relevant explanation will be added in the manuscript. 13. Line 277-278 What is the meaning of "Because all these vegetation types are more or less affected by human activities, soil erodibility can also indirectly be affected by vegetation recovery and land cover change."? Please rewrite it to make it easy to be understand. Reply: We are sorry for causing this confusion. This sentence indicated that human activities affect the vegetation recovery and land cover change, then change the vegetation types and these changes may influence the soil erodibility. This sentence will be rewritten to make it easier to read.

---

## Author Comment (AC2) · 2 Aug 2018

Dear reviewer #2, Thank you very much for your comments and suggestions. The comments inspired us to think more about the details of the analysis and the correction of explanation such as abbreviations and annotations. The following is a point-by-point response to all of the comments. Once again, we appreciate for your kind work, and hope that the correction will meet with approval. Best regards,

Wenwu Zhao Email: zhaoww@bnu.edu.cn

Point-by-point responses: 1. Abstract. Hardly any quantitative result is found in the ab-

stract. Reply: The main conclusions of this paper are (1) Shirazi and Torri model was considered as the optimal models for Ansai watershed; (2) since soil erodibility is estimated by soil properties, soil erodibility has an indirect relationship with environmental factors, including elevation and slope degree, and to a lesser extent, human activities; (3) by changing vegetation density, biomass, and cover, human can indirectly affect soil erodibility. The abstract will be revised by clearly express the quantitative results of the study. 2. Results. Sections 3.2 and 3.3: where is Tables S1–S5? I try to find the relationships of the text with Tables 1-4, but I am failed. I am sure some tables have been lost in the manuscript. Reply: We feel so sorry for this. The Tables S1–S5 (also including S6) will be further submitted as Supporting Information. The clear citation of these tables will be added in the manuscript and this Supporting Information will also be noted to this comment. 3. Lines220-222, Page 11: Table 3 in page 22 presents the Principal component analysis (PCA) of environmental attributes, instead of the MDS of the soil erodibility. Reply: We are sorry to make this mistake. This should be the S1-S3 in the Supporting Information. This mistake will be corrected and all the citation of tables and figures will be checked. 4. Some of the tables have been published in a Chinese journal. For example, Figure 2 in the manuscript is similar to Figure 3 in Reference (Zhao et al., 2017). I have uploaded the published paper together with the comments. Reply: We are sorry for causing this confusion. Some parts of these two papers used the same group of data which was obtained from field work. But the topic of these data and analysis are totally different in these two papers. The figure 2 in the manussript will be removed and relevant discussion will be added to show the differences between these two papers. 5. Too many abbreviations have been found in the manuscript. I think you may make a list for the abbreviations as an accessory of the paper. Moreover, some of the abbreviations are not needed, e.g., the words skewness and kurtosis in Table 2. Reply: A list of the abbreviations will be added as an accessory of the paper and the abbreviations that are not needed will be removed. 6. Errors exist in the annotations. Some of the annotations followed with the tables are duplicated, e.g., the annotations in Tables 1 and 3. I suggest the parameters s in the

may be emerge according to their order in the table. I am sorry I could not find SP and SS in Table 1, although the terms have been explained in the annotation. Reply: All the annotations will be checked and corrected by emerging them according to their order in the table and ensuring all the abbreviations and labels are noted in the annotations. 7. English writing of the manuscript is readable. Nevertheless many language errors exist. I strongly suggest you ask a soil scientist whose native language is English to polish the whole manuscript. Reply: We will check our manuscript carefully and correct the errors. Then the manuscript will be polished by soil scientist whose native language is English.

---

## Author Response (AR1)

Dear Reviewer #1,

Thank you very much for the constructive comments and suggestions. Please find below our point-by-point responses to all of the comments. We have added detailed discussions and corrected the language errors as you suggested. The page and line numbers in the following responses refer to the revised manuscript "Manuscript without annotation.". We appreciate the time that you spent reviewing the manuscript.

Best regards,

Wenwu Zhao

Email: zhaoww@bnu.edu.cn

**Point-by-point responses:**

1.  It is better to polish the language further.

    Response: The language has been carefully checked.

2.  Line 30-31, Please delete the detail soil properties in bracket "(e.g., soil texture, permeability and structural stability)".

    Response: The detailed soil properties in parentheses have been deleted. Please see line 30.

3.  Line 34 Change "research" to "researches"

    Response: The word "research" has been changed to "researches". Please see line 33.

4.  Line 40 Abbreviations should be added in the following of "the nomogram model and the modified nomogram model"

    Response: Acronyms have been added following "the nomogram model and the modified nomogram model". The sentence has been revised to "Some of the most common estimation models are the nomogram model (NOMO) and the modified nomogram model (M-NOMO), which were established by Wischmeier". Please see lines 39-40.

5.  Line 86-87 references about the classification of soil particles should be added.

    Response: We have cited Wang et al. 2012 (Wang, D. C., Zhang, G. L., Pan, X. Z., Zhao, Y. G., Zhao, M. S., Wang, G. F.: Mapping soil texture of a plain area using fuzzy-c-means clustering method based on land surface diurnal temperature difference, Pedosphere, 22, 394-403, 2012) accordingly. Please see line 86 and lines 342-344.

6. Line 91-92 what is the meaning of "Soil erodibility thus has indirect relationship with the environmental factors." ? Please rewrite it.

Response: The intended meaning of this sentence is that although environmental factors may not have direct influences on soil erodibility, they can affect soil erodibility by affecting soil particle size distribution and soil organic matter content. The text has been rewritten for clarity as follows: "Although soil erodibility does not directly depend on environmental factors, soil properties such as soil particle size distribution and soil organic matter can be affected by environmental factors; thus, environmental factors have indirect relationships with soil erodibility". Please see lines 89-91.

7. Line 106 "rainfall erosion" or "rainfall erosivity"?

Response: The text "rainfall erosion" has been changed to "rainfall erosivity". Please see line 105.

8. Line 154 Please change "P value >0.05" into "P > 0.05".

Response: The text "P value >0.05" has been changed to "P > 0.05". Please see line 152.

9. Part 3.2 Significant negative or positive correlations are in P < 0.05 or P < 0.01? Need be labelled in the following.

Response: The significance of each correlation (P < 0.05) has been added in section 3.2. Please see lines 179, 181, 183, 186, 190, 192, 193 and 195.

10. Line 202-203 One PC each for apple orchards, native grasslands, sea buckthorn, Caragana korshinskii and pasture grasslands. Why there is only 4 data of percentage in the following sentence. Please check it.

Response: Pasture grasslands had no variables with high factor loadings because it had no significant environmental variables except soil particle size and soil organic matter. Therefore, there are only 4 data of percentage listed. This information has been added to the revised manuscript. Please see lines 201-202.

11. Line 264 Soil erodibility has significant correlations with elevation? Please check it. If so, explain why.

Response: Soil erodibility showed significant correlations with elevation in this study. According to previous studies, elevation in certain areas might have relationships with factors such as soil and vegetation type; both had significant correlations with soil erodibility in this study. We have added the following to the manuscript accordingly: "Terrain factors have close relationships with soil properties. With changes of elevation and slope, the physical and chemical properties of soil (e.g., soil permeability, soil bulk density, and soil nutrients) and soil surface conditions (e.g., roughness, litter layer) change, leading to changes in soil particle size composition and soil erodibility..." Please see lines 258-261.

12. Line 267 What "soil surface conditions" refer to? Please give some examples.

Response: Soil surface conditions are the result of many factors such as surface roughness and vegetation coverage. We have added "soil surface conditions (e.g., roughness, litter layer)" to the manuscript. Please see lines 260.

13. Line 277-278 What is the meaning of "Because all these vegetation types are more or less affected by human activities, soil erodibility can also indirectly be affected by vegetation recovery and land cover change."? Please rewrite it to make it easy to be understand.

Response: We apologize for the unclear writing; we have revised the text for clarity. The sentences state that human activities affect vegetation recovery and land cover change and that changes in vegetation type may influence soil erodibility. The revised text is as follows: "Human activities (e.g., pruning) affect vegetation recovery and land cover change. These changes may then influence vegetation properties and thereby impact soil erodibility." Please see lines 271-272.

**Other changes:**

We have revised the manuscript carefully and made some additional changes (especially regarding English language issues) in the manuscript. These changes do not influence the content or framework of the paper. All the changes to the manuscript are marked by the track changes feature of Word.

Dear Reviewer #2,

We thank you for the constructive comments and suggestions. Please find below our point-by-point responses to all of the comments. We have ensured that all abbreviation, tables and figures are accurately referenced in the manuscript, and we have corrected the language errors as you suggested. The page and line numbers in the following responses refer to the revised manuscript "Manuscript without annotation". We appreciate the time that you spent reviewing the manuscript.

Best regards,

Wenwu  Zhao

Email: zhaoww@bnu.edu.cn

**Point-by-point responses:**

1. Abstract. Hardly any quantitative result is found in the abstract.

    Reply: We thank you for the comment. We have added the following quantitative results to the abstract: "(1) $K$ values in the Ansai watershed ranged between 0.009 and 0.092 t·hm$^2$·hr/(MJ·mm·hm$^2$), and the maximum values were 1.872-7.333 times larger than the corresponding minimum values, and the Shirazi and Torri models were considered the optimal models for the Ansai watershed. (2) Different land use types had different levels of importance; PC one accounted for 100% (native grassland), 48.88% (sea buckthorn), 62.05% (*Caragana korshinskii*) and 53.61% (pasture grassland) of the variance in soil erodibility." Please see lines 14-19.

2. Results. Sections 3.2 and 3.3: where is Tables S1–S5? I try to find the relationships of the text with Tables 1-4, but I am failed. I am sure some tables have been lost in the manuscript.

    Reply: We apologize for the oversight. Tables S1–S5 and the new Table S6 have been submitted as Supporting Information. In addition, we have referenced these tables clearly in the manuscript as you suggested. Please see the file named Supporting Information.

3. Lines220-222, Page 11: Table 3 in page 22 presents the Principal component analysis (PCA) of environmental attributes, instead of the MDS of the soil erodibility.

    Reply: We apologize for the oversight. We have corrected the information in Table S1-S3 in the Supporting Information. We also have checked all of the references to tables and figures in the manuscript. Please see the file named Supporting Information.

4. Some of the tables have been published in a Chinese journal. For example, Figure 2 in the manuscript is similar to Figure 3 in Reference (Zhao et al., 2017). I have uploaded the published paper together with the comments.

Reply: The reference you mention (Zhao et al., 2017) is "Wei, H., Zhao, W. W., Wang, J.: The optimal estimation method for K value of soil erodibility: A case study in Ansai Watershed, Science of Soil and Water Conservation, 15, 52-62, 2017b. (in Chinese with English abstract)". We acknowledge that this paper and our manuscript use the same data on soil erodibility. However, the topics and analyses fully differ between these two papers. We have removed Fig. 2 from the current manuscript and have cited the previous paper when mentioning these data. The relevant sentence has been revised to "To clarify the form of the distribution, we collected the frequency distribution figures of soil erodibility for each model (Wei et al., 2017a, b)." Please see lines 150-151.

5. Too many abbreviations have been found in the manuscript. I think you may make a list for the abbreviations as an accessory of the paper. Moreover, some of the abbreviations are not needed, e.g., the words skewness and kurtosis in Table 2.

Reply: A list of abbreviations has been added, and unnecessary abbreviations have been removed from the manuscript. The abbreviations in the list are presented in alphabetical order. Please see List 1 in the file named Supporting Information.

6. Errors exist in the annotations. Some of the annotations followed with the tables are duplicated, e.g., the annotations in Tables 1 and 3. I suggest the parameters in the may be emerge according to their order in the table. I am sorry I could not find SP and SS in Table 1, although the terms have been explained in the annotation.

Reply: We apologize for the oversight. SP and SS have been deleted from the annotation of Table 1. All of the annotations have been reviewed and corrected according to their order in the table to ensure that all of the abbreviations and labels are addressed in the annotations. Please see lines 398-417.

7. English writing of the manuscript is readable. Nevertheless, many language errors exist. I strongly suggest you ask a soil scientist whose native language is English to polish the whole manuscript.

Reply: We thank you for the suggestion, and we have checked the manuscript carefully and corrected the errors.

**Other changes:**

We have revised the manuscript carefully and made some additional changes (especially regarding English language issues) in the manuscript. These changes do not influence the content or framework of the paper. All the changes to the manuscript are marked by the track changes feature in Word.